# The Effects of Nanoparticles- Zerovalent Iron on Sustainable Biomethane Production through Co-Digestion of Olive Mill Wastewater and Chicken Manure

Khalideh Al Bkoor Alrawashdeh [1,*], Kamel K. Al-Zboon [2], Jalal A. Al-Tabbal [3], La'aly A. AL-Samrraie [2], Abeer Al Bsoul [4,*], Rebhi A. Damseh [1], Ayat Khasawneh [2], Yasser Dessouky [5], Kareem Tonbol [6], Bassma M. Ali [7] and Elen E. Youssef [7]

[1] Mechanical Engineering Department, Al-Huson University College, Al-Balqa Applied University, P.O. Box 50, Al-Huson, Irbid 19117, Jordan
[2] Water and Environmental Engineering Department, Al-Balqa Applied University, P.O. Box 50, Al-Huson, Irbid 19117, Jordan
[3] Department of Nutrition and Food Processing, Al-Huson University College, Al-Balqa Applied University, P.O. Box 50, Al-Huson, Irbid 21510, Jordan
[4] Department of Chemical Engineering, Al-Huson University College, Al-Balqa Applied University, P.O. Box 50, Al-Huson, Irbid 19117, Jordan
[5] College of Engineering and Technology, Arab Academy for Science, Technology and Maritime Transport, Abu-Qir P.O. Box 1029, Egypt
[6] College of Maritime Transport and Technology, Arab Academy for Science, Technology and Maritime Transport, Abu-Qir P.O. Box 1029, Egypt
[7] College of Pharmacy, Arab Academy for Science, Technology and Maritime Transport, Abu-Qir P.O. Box 1029, Egypt
* Correspondence: khalideh19@bau.edu.jo (K.A.B.A.); dr.abeeralbsoul@bau.edu.jo (A.A.B.)

**Abstract:** The impacts of nanoparticles-zerovalent iron (NP-ZVI) on anaerobic co-digestion (AcoD) were assessed. The production of biogas and methane ($CH_4$), as well as the removal efficiency of volatile solids (VS) and contaminants were investigated in the AcoD of chicken manure (CM) and olive mill wastewater (OMWW) with the addition of NP-ZVI at different concentrations (10–50 mg/g VS) and different sizes resulting from various mixing volume ratios (MVR) of $NaBH_4$:$FeSO_4$.$7H_2O$. The results show that NP-ZVI $\leq$ 30 mg/g VS at MVR-2:1, MVR-4:1, and MVR-6:1 improves the AcoD. In contrast to 40–50 mg/g VS of NP-ZVI, which caused an inhibitory impact in all of the AcoD stages, as well as a decrease in the contaminant's removal efficiency, the concentration of 10–30 mg NP-ZVI/g VS at MVR-4:1 achieved a maximum improvement of $CH_4$ by 21.09%, 20.32%, and 22.87%, respectively, and improved the biogas by 48.14%, 55.0%, and 80.09%, respectively, vs. the 0 additives. Supplementing AcoD with NP-ZVI at a concentration of 30 mg/g VS at MVR-4:1 resulted in maximum enhancement of the contaminant removal efficiency, with a total oxygen demand (TCOD) of up to 73.99%, turbidity up to 79.07%, color up to 53.41%, total solid (TS) up to 59.57%, and volatile solid (VS) up to 74.42%. It also improved the hydrolysis and acidification percentages by up to 86.67% and 51.3%, respectively.

**Keywords:** anaerobic co-digestion process; zerovalent iron nanoparticles; olive mill waste; COD removal; BOD removal; biogas production

## 1. Introduction

The anaerobic digestion (AD) technique has been developed for processing a broad range of organic wastes, including olive mill wastewater (OMWW) and chicken manure (CM), as the need for renewable energy and waste management rules that the mandated recycling of wastes continue to grow [1–5]. OMWW and CM have significant levels of organic matter, which calls attention to them as environmentally beneficial materials for nutrient recovery and the water-energy nexus [6]. The anaerobic co-digestion (AcoD)

of OMWW and CM is considered promising for improving biogas production as this process overcomes the lack of nutrients in OMWW, reduces the pollutant (e.g., TCOD) concentrations, obtains an ideal C/N ratio, and improves the stability, as required by the AD process [7]. The AcoD of OMWW and CM reduces the input organic matter's chemical oxygen demand (COD) and biochemical oxygen demand (BOD), which it uses for energy recovery in the form of biogas [8–11]. Additionally, it was reported that the AcoD of OMWW and CM enhanced biogas generation by 22% [12] and volatile solids (VS) and total solids (TS) removal by 22% and 32%, respectively [11,12].

Previous research has evaluated and recommended creative methods for improving AcoD's effectiveness. Supplemental co-digestion, as part of pretreatment, is one of these approaches. Nanoparticulate iron compounds (Fe) are one of the additions that can mitigate ammonia inhibition and improve biogas generation [13]. The addition of zerovalent iron (ZVI) to AcoD systems has been associated with increasing the efficiency of hydrolysis, acidogenesis and methane ($CH_4$) production; preventing excessive acidification; enhancing COD removal; and improving the conversion of propionate to acetate [14,15]. According to previous studies, a variety of enzymatic reactions require ZVI as a cofactor, promoting fermentation and creating more favorable conditions for AcoD due to its capacity to reduce the oxidative-reductive potential of the digestion medium [16].

Some studies have mentioned the potential of ZVI. According to Deng et al. [17], ZVI improves autotrophic denitrification during the AD process. In addition, an investigation study conducted by Tchobanoglous [18] showed that phosphate removal could be realized through adsorption by hydrous ferric, phosphate incorporation into the structure of hydrous oxide, and the formation of ferric phosphates that are removed as precipitates. In addition to ZVI's inherent properties, the AcoD tests the operating factors, such as temperature, ZVI concentration, active volume, pH, mixing setting, and contact duration. In addition, dissolved oxygen impacts how ZVI performs in removing contaminants [19–21].

The processes and phases of the AD system (hydrolysis, acidogenesis, acetogenesis, and methanogenesis) for different types of organic wastes have been the subject of several investigations on ZVI-supported AD [13,22,23]. However, each ZVI material has a unique inherent reactivity. As a result, varied performance is anticipated in ZVI-supported AD systems using biowaste, which may perform differently at varying concentrations of ZVI [24–26]. The available results, however, were also obtained from studies carried out in significantly diverse conditions. This made it difficult to assess the importance of the discrepancies based only on those results.

Most research studies have focused on the effects of size on the complement's ability to remove or reduce a wide variety of contaminants in recent years [13,22]. The use of nanoparticles (NP) has beneficial impacts on the AD process, notably regarding improving the yield and rate of methane generation, reducing the startup and recovery times, and improving the stability. In contrast to their bigger predecessors, the resulting particles are extremely reactive and suitable for the in-situ treatment of various pollutants. Most pollutants can be eliminated or stabilized by NP-ZVI [26].

Additionally, nanoscale ZVI provides a larger surface area with increased surface activity compared to bulk or macro-scale ZVI. Zero-valent iron nanoparticles (NP-ZVI) are small enough to be injected and distributed easily in porous media. According to Lizama et al. [27], the impact of NP-ZVI on $CH_4$ production was assessed, and the study results showed an increase in the rate and yield of $CH_4$. This enhancement is attributed to the possibility of iron contributing as an electron donor when hydrogenotrophic methanogens directly reduce $CO_2$ to $CH_4$. On the other hand, Yang et al. [28] found that NP-ZVI (55 nm), corresponding to 56 mg/L (1 mM), reduced the methanogenesis of the AD process by 20%. The investigation demonstrated that the disruption of the cell membranes caused by NP-ZVI and the resulting rapid hydrogen generation as a result of NP-ZVI dissolution inhibited methanogenesis [23]. A variety of methanogens can use NP-ZVI as the only electron donor. Methanogens can obtain electrons from NP-ZVI more efficiently than hydrogen utilization [29]. The potential benefits of NP-ZVI on AcoD include the elimination of $H_2S$,

the stimulation of essential enzymes involved in the processes of acidification and methanogenesis, and a decrease in the oxidation-reduction potential (ORP) [30]. For AD, a low ORP environment may be more favorable. This also involves the directed flow of electrons [26]. Contaminants obtain the electrons produced by ZVI, transforming them into less toxic or non-toxic compounds. The environment-generated electrons from ZVI corrosion assist microorganisms' metabolic processes. By donating electrons to $O_2$, dissolved oxygen is used to produce $H_2O_2$ [31–34].

Despite the above-mentioned positive effects, using NP-ZVI at excessive dosages can have inhibitory consequences. These inhibitory effects can be related to the highly catalytic environment produced at the NP-ZVIs surface, which can instantly suppress bacterial activity by severely damaging the membrane surface and metabolic processes through the reductive degradation of enzyme functional groups [35,36], as well as, presumably, to the accelerated hydrogen synthesis and concentration that led to the accumulation of volatile fatty acids (VFAs) [37]. The level of promoting and inhibiting NP-ZVI supplemental doses during the AcoD process was not identified in the studies that assessed NP-ZVI at specific concentrations.

Therefore, this study aimed to investigate the performance of the co-digestion of OMWW and CM with the addition of NP-ZVI in various concentrations (10–50 mg/g VS) and sizes. Various sizes of NP-ZVI were synthesized by preparing different mixing volume ratios of $NaBH_4$:FeSO4.7$H_2O$ (2:1, 4:1, 6:1, 8:1, and 10:1). The impacts of the concentration and size of NP-ZVI were investigated in terms of the biogas yield, the content of $CH_4$ (%), the turbidity, the color, the AcoD process stability, and the removal efficiencies of VS, TS, TCOD.

## 2. Materials and Methods

### 2.1. Substrates

The substrate of CM was collected from five poultry farms located in north Irbid, Jordan. Meanwhile, the OMWW substrate was obtained from the three-phase oil extraction plant immediately after the harvest season of 2022. The substrates of CM and OMWW were stored at four °C until used. The inoculum used for the co-digestion of CM and OMWW waste was obtained from previous AD tests [13].

### 2.2. Analytical Methods

The substrates of OMWW, CM, and inoculum, as described by Alrawashdeh et al. [2], were prepared to be used in the AD tests in accordance with UNI 5667-13/2000. The substrate characteristics were analyzed using a thermogravimetric analyzer (TGA 701, LECO, St. Joseph, MI, USA) based on the physicochemical and proximate characteristics. The analyses included volatile solids (VS), total solids (TS), humidity (U), ASH, and fixed carbon (FC), as reported in [5]. The loss of volatile substances, including VFAs, can result in mass errors. An incineration at 550 °C removes all organic matter from the sample to determine its ash content. After burning off the VS portion, only the ash remains. The VS can be calculated from TS minus the ash content.

The ultimate total carbon, hydrogen, and nitrogen analysis were performed using a Leco TruSpec CHN analyzer (LECO CR-412, St. Joseph, MI, USA) in accordance with ASTM D5373 [38,39].

The content of Fe was determined using atomic absorption spectroscopy (novAA 800, Jena, Germany), and the standard method 4500-NH3 was used to measure the ammonia level. To evaluate the original nitrate present in the sample, 2,6-dimethylphenol is also reacted with the undigested sample [40,41]. Then, using Equation (1), TKN is

$$TKN = TN - \text{Nitrate and Nitrite} \qquad (1)$$

The COD was measured using the APHA standard methods [42] for both soluble and total COD. The TCOD was determined by a thermo-reactor (AL125-AQUALYTIC, Nairobi, Kenya). The turbidity was measured with a turbidity and chlorine portable meter

(Model HI93414-02). Utilizing a Hach DR/4000 spectrophotometer (APHA, 1989), the phosphorous content is determined by vanadomolybdophosphoric acid colorimetry. Gas chromatography (GC) was performed to quantify the VFA, according to Beo et al. [43].

Scanning electron microscopy (SEM) was used to characterize the produced nanoparticles. Using an FEI-Quanta 200F (Thermo Fisher, Waltham, MA, USA) machine, a scanning electron microscope was used to characterize the nanoparticle morphology and mean particle size. The Brunauer Emmett-Teller (BET) adsorption isotherm method was used to calculate the specific surface area of the nanoparticles, as reported by Ahuja et al. [44].

The Agilent 490 Micro gas chromatograph (Agilent Technologies Inc., Santa Clara, CA, USA) was used to analyze the biogas samples [5,6]. To avoid pressurized conditions and explosion threats, extra biogas was regularly vented. A portable pH meter (Model HI9819) with double junction pH electrodes was used to measure the pH of the samples [10].

### 2.3. The Hydrolysis and Acidification Rate, and VS Destruction

The total solubilized products (equivalent of $CH_4$ produced and $SCOD$) and particle $COD$ concentrations in the substrate mixture were to assess the hydrolysis percentage of the substrate (solubilization) [21,43]. Depending on the rate of hydrolysis, we can express it as follows:

$$Hydrolysis\ (\%) = \frac{COD_{CH4} + SCOD_o - SCOD_i}{TCOD_i - SCOD_i} \times 100\% \qquad (2)$$

Where $COD_{CH4}$ is the amount of methane produced at a specific time during the co-digestion process and is expressed as mg COD. $SCOD_i$ and $SCOD_o$ are the concentrations of soluble COD at the initial substrate and at the discharge, respectively, and $TCOD_i$ is the total COD concentration at the initial substrate.

The acidification is related to VFA production, the loading rate of $COD$, and double-bond removal. The acidification rate is essential in evaluating the digestion process's efficiency [45–48]. Equation (3) was used to evaluate the acidification percentage.

$$Acidification\ (\%) = \frac{COD_{CH4} + COD_{VFA} - COD_{VFA,i}}{COD_{in}} \times 100\% \qquad (3)$$

where $COD_{VFA}$ is the VFA equivalent COD at any time during the digestion process, $COD_{VFA,i}$ is the VFA equivalent COD at the beginning of digestion, and $COD_{in}$ is the initial COD concentration.

### 2.4. Nanoparticles- Zerovalent Iron Preparation

The synthesis of NP-ZVI was prepared by a chemical reduction method, according to Ahuja et al. [44]. Sodium borohydride ($NaBH_4$) was used to reduce the ferrous sulfate ($FeSO_4.7H_2O$) in an aqueous mixture. A solution of 0.05 M $FeSO_4.7H_2O$ was reduced by using 0.2 M $NaBH_4$. $NaBH_4$ was slowly added into $FeSO_4.7H_2O$ under continuous mixing by a magnetic stirrer (400 rpm) at room temperature. The mixing volume was set to 4:1 for faster particle reduction. Both the pH and temperature were kept constant in the ambient conditions to detect the oxidation impact on iron. The chemical reaction involves the synthesis of suitable nanoparticles with the various ratios of mixing volume (MVR) of 0.05 M $Fe_2SO_4.7H_2O$ (100 mL) and 0.2 M $NaBH_4$ (25 mL), as described in Equation (4). Various MVRs (MVR-2:1, MVR-4:1, MVR-6:1, MVR-8:1, MVR-10:1) were conducted to create different sizes and specific surface areas.

$$4Fe^{3+}_{(aq)} + 3BH_4 + 9H_2O \rightarrow 4Fe^0_{(s)} + 3H_2BO_3 + 12H+_{(aq)} + 6H_{2(g)} \qquad (4)$$

The reduction reaction generates colloidal precipitates, which cause the color to instantly shift from clear brownish to black. The solution (containing nanoparticles of iron) was continuously mixed for 5 min, after which the particles were separated using centrifugation (3800 rpm) for 3–4 min. Following separation, the precipitate was washed

three times with distilled water to remove excess $NaBH_4$ and then washed four times with ethanol. The precipitate was moved to an ethanol medium to prevent it from oxidizing, and finally, it was dried at 600 °C in an electric oven (for 8 h).

Nanoparticles of about 20–100 nm size, mostly spherical and chain-structured, were synthesized in a laboratory. The samples of MVR −4:1 had the smaller mean crystallite size among all the samples. The size of the samples was classified by MVR-4:1 (20–56 nm) < MVR-2:1 (24.7–65 nm) < MVR-6:1 (31–60 nm) < MVR-8:1 (43.5–89.2 nm) < MVR-10:1 (47.6–100 nm).

The NP-ZVI was prepared in different concentrations for each mixing volume ratio (MVR), which were added to BMP tests, as reported in Table 1.

**Table 1.** Symbols of the BMP tests that were supplemented with various concentrations of NP-ZVI which were obtained at various MVRs.

| Mixing Volume Ratio ($NaBH_4:FeSO_4.7H_2O$) | mg NP-ZVI/ g VS | | | | |
|---|---|---|---|---|---|
| | **10** | **20** | **30** | **40** | **50** |
| **2:1** | $A_{10}$ | $A_{20}$ | $A_{30}$ | $A_{40}$ | $A_{50}$ |
| **4:1** | $B_{10}$ | $B_{20}$ | $B_{30}$ | $B_{40}$ | $B_{50}$ |
| **6:1** | $C_{10}$ | $C_{20}$ | $C_{30}$ | $C_{40}$ | $C_{50}$ |
| **8:1** | $D_{10}$ | $D_{20}$ | $D_{30}$ | $D_{40}$ | $D_{50}$ |
| **10:1** | $E_{10}$ | $E_{20}$ | $E_{30}$ | $E_{40}$ | $E_{50}$ |

*2.5. Experimental Setup*

In accordance with the preferred moisture content (90%) and desired C:N ratio required by the AcoD process, the substrate mixture content was determined to be 78:22 *v/v* of OMWW:CM, improving the nutritional balance for microbial growth and enhancing the buffering capacity for stable AcoD [9,13,20].

The BMP test was carried out in a 1 L vessel with a working volume of 200 mL. The vessel's outlet is equipped with two holes, both used for collecting samples, controlling the pH values, and allowing the generated biogas to pass through. The ratio of feedstock-to-inoculum was 90:10 *v/v* (the feedstock is a mixture of OMWW and CM). The vessels were flushed with nitrogen ($N_2$) gas to remove air and oxygen and were tightly sealed; then, they were immersed in a water bath to maintain the mesophilic conditions (approximately 40 ± 0.2 °C). Biogas production was collected by attaching a gas bag (1 L) equipped with a double-valved Luer adapter for sampling and an elastic tube to each vessel. The biogas production was calculated daily using a thermocouple and a pressure gauge attached to each bag. According to Reference [47], each vessel was manually shaken twice daily (1 min) during the tests.

BMP tests were conducted with NP-ZVI concentrations of 10, 20, 30, 40, and 50 mg NP-ZVI/g VS. The effect of the NP-ZVI doses on the AcoD process was investigated and identified in order to specify the optimum concentration of NP-ZVI for improving the AcoD process. Different factors were investigated to detect the optimum concentration of NP-ZVI in terms of the process stability, digestion retention time, methane production, and biogas production, as well as the removal efficiencies of TCOD, color, turbidity, TS, and VS.

The pH value was measured daily, and due to the high acidity of the substrate mixture, during the initial stage, it was adjusted to approximately seven every three days by adding 1.2 mL of KOH (90% of concentration) to each vessel.

After adding the designed concentrations of NP-ZVI to the substrate mixture in the batch digesters, the BMP reactors began to run and required 30 days to reach stability. A test with inoculum alone was used as a blank, and the biogas produced by the blank was subtracted from the BMP tests. To detect the effect of the addition of NP-ZVI on the AcoD, control tests using the substrate mixture alone, without NP-ZVI addition, were also conducted.

## 3. Results

Table 2 shows the characteristics of OMWW, CM, and inoculum as a single substrate. The substrate mixture has a VS/TS of 28.96%, which lies within the recommended range reported in the literature [13,27–29]; this indicates its suitability as a raw material in the AcoD process and its high potential to convert the organic fraction into biogas. In addition, the substrate mixture is characterized by a high content of contaminants; this can slow down degradation because hydrolysis will be limited, where the TCOD, $BOD_5$, and phenols are $41.645 \pm 2.03$ mg/L, $7.705 \pm 0.41$ mg/L, and $6.858 \pm 0.38$ mg/L, respectively.

**Table 2.** The OMWW, CM, and the inoculum characteristics.

| Substrate | Total Solids, TS (%) | Volatile Solids, VS (%) | Ash (%) | Moisture (%) | Fixed Carbone (%) | Phenols (gl $^{-1}$) | TCOD (gl $^{-1}$) | $BOD_5$ (gl $^{-1}$) | pH |
|---|---|---|---|---|---|---|---|---|---|
| OMWW | $7.12 \pm 0.96$ | $2.92 \pm 0.52$ | $0.39 \pm 0.09$ | $92.88 \pm 0.97$ | $3.81 \pm 0.8$ | $4.21 \pm 0.15$ | $53.79 \pm 2.04$ | $8.43 \pm 3.56$ | $5.08 \pm 1.32$ |
| CM | $27.65 \pm 2.82$ | $5.12 \pm 1.02$ | $2.31 \pm 0.21$ | $72.35 \pm 2.05$ | $20.22 \pm 2.7$ | $6.83 \pm 0.65$ | $9.05 \pm 1.96$ | $7.21 \pm 1.08$ | $6.39 \pm 0.18$ |
| Inoculum | $2.51 \pm 0.62$ | $0.53 \pm 0.06$ | $0.51 \pm 0.07$ | $97.49 \pm 1.04$ | $1.47 \pm 0.2$ | $2.95 \pm 1.36$ | $21.82 \pm 2.91$ | $3.62 \pm 2.13$ | $6.75 \pm 0.95$ |

The C:N ratio of the OMWW, CM, inoculum, and substrate mixture was 34.96, 9.08, 17.23, and 28.011, respectively. Several researchers have demonstrated that the optimal C/N ratios range between 25–30 [35,38], where the microorganisms consume C in the range of 25–30 times faster than N [47]. The Fe concentration of the OMWW, CM, inoculum, and substrate mixture was $30.43 \pm 1.25$, $132.45 \pm 4.19$, $23.87 \pm 1.63$, and $50.178 \pm 2.17$ mg/L, respectively. The synthetic substrate mixture has a COD:N:P ratio of 230:5:1. Hence, in the nutrient supplementation context, according to the literature, the recommended ratio is 250:5:1 [49,50], indicating that the substrate mixture is suitable for the AcoD process. To reach stability, the tests were conducted for roughly 30 days; the pH dropped sharply at startup, reaching extreme acid levels for each test. As mentioned before, the pH adjustments improved the digestion rates.

### 3.1. Influence of NP-ZVIs on the AcoD Process

The features of iron (such as particle size and oxide layers), the operational parameters (such as pH value and iron dosage), and the other variables all have an impact on the effect of NP-ZVI in the AcoD process. The AcoD performance is discussed in this section in relation to the pretreatment dosage and particle size.

### 3.1.1. Effects of Dose and Size of NP-ZVIs on the Hydrolysis Percentage

The improved rate and efficiency of the hydrolysis performance of the NP-ZVI addition were directly correlated to its concentration and particle size. The strengthening effect on the kinetics of AcoD is more pronounced the smaller the particle size because the rate of solid organic conversion into soluble is higher. The results showed that the AcoD hydrolysis was accelerated by the increased surface area of NP-ZVI. NP-ZVI dosages of 10–50 mg NP-ZVI/g VSco-substrate at different MVRs of NaBH4:FeSO4.7H2O, as reported in Table 1, were added to the AcoD tests. Figure 1a–c shows the effects of adding NP-ZVIs (different concentrations and MVRs) on the concentration of SCOD and the hydrolysis percentage, which represent the transformation of TCOD into SCOD after one day, six days, and fifteen days, respectively. The results show that the lowest reduction of TCOD into SCOD was observed when the NP-ZVI was significantly higher than 30 mg/g VS. The highest concentration of SCOD was achieved by test B30 (30 mg NP-ZVI/g VS at MVR-4:1). As shown in Figure 1a–c, the addition of test B30, versus the control test, increased the SCOD concentrations by 30.9%, 43.6%, and 31.31% on day one, day six, and day fifteen, respectively. According to Table 3, versus the control test, the maximum hydrolysis (%) was observed at MVR-4:1 for 30, 20, 10, and 40 mg NP-ZVI/g VS with an improvement of 86.67%, 74.89%, 74.33%, 63.45%, respectively. Meanwhile, 50 mg NP-ZVI/g VS achieved an improvement by 65.15 at MVR-6:1. Where the SCOD concentration of the control test

was 1.78, 2.5, and 0.99 g/L after day one, day six, and day fifteen, respectively, and a 44.1% hydrolysis percentage.

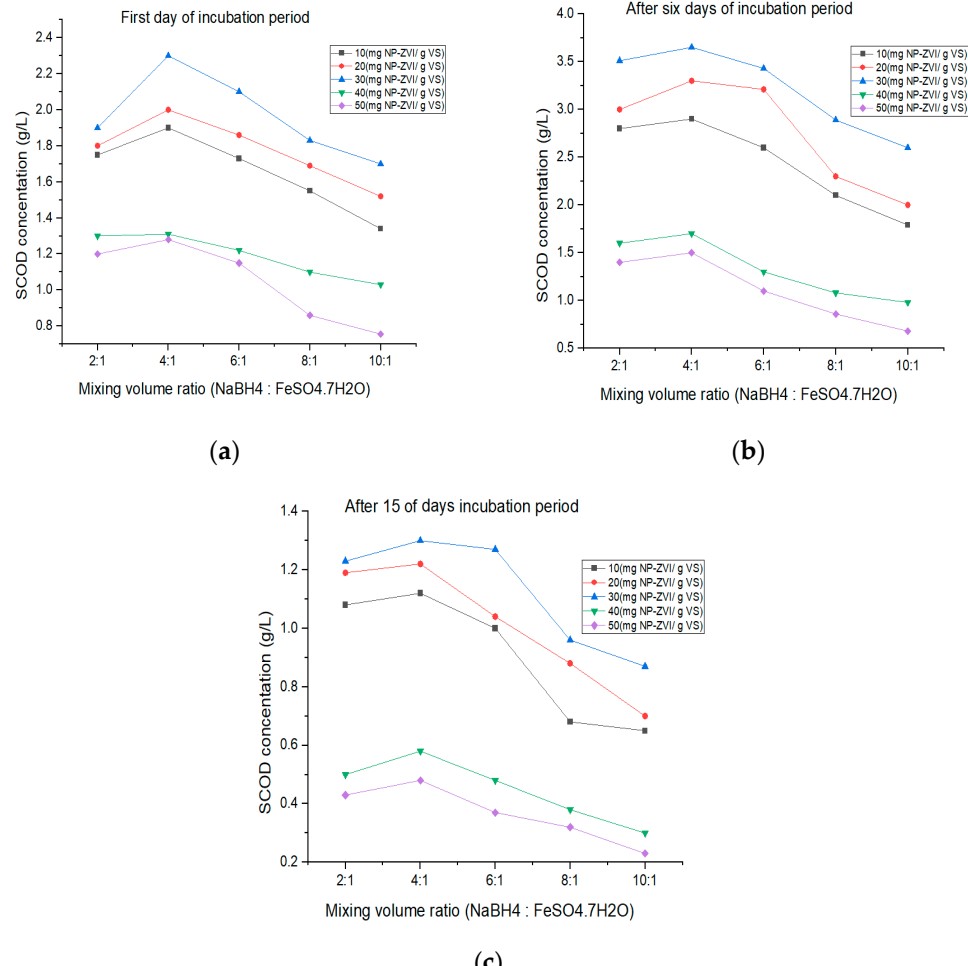

**Figure 1.** (**a**) Effect of NP-ZVI doses and various MVR on the concentration of SCOD after one day of the incubation period, where the SCOD concentration of the control test was 1.78 g/L; (**b**) Effect of NP-ZVI doses and various MVR on the concentration of SCOD after six days of the incubation period, at the same time the SCOD concentration of the control test was 2.5 g/L; (**c**) Effect of NP-ZVI doses and various MVR on the concentration of SCOD after 15 days of the incubation period, at the same time the SCOD concentration of the control test was 0.99 g/L.

**Table 3.** The hydrolysis percentage of various NP-ZVI doses and mixing volume ratios ($NaBH_4$:$FeSO_4.7H_2O$).

| Mixing Volume Ratio ($NaBH_4$:$FeSO_4.7H_2O$) | Hydrolysis (%) | | | | |
|---|---|---|---|---|---|
| | **10 mg/g VS** | **20 mg/g VS** | **30 mg/g VS** | **40 mg/g VS** | **50 mg/g VS** |
| 2:1 | 75.25 | 73.9 | 74.13 | 69.84 | 65.18 |
| 4:1 | 76.88 | 77.13 | 82.32 | 72.08 | 71.77 |
| 6:1 | 73.52 | 74.37 | 78.60 | 65.93 | 72.83 |
| 8:1 | 61.63 | 58.75 | 63.52 | 60.39 | 60.82 |
| 10:1 | 54.61 | 56.91 | 57.26 | 57.67 | 64.65 |

The peak of the SCOD concentration (TCOD transformation) was reached after the first six days for all of the test incubations of the NP-ZVI doses. The peak of the SCOD concentration (TCOD transformation) was reached after the first six days for all the test incubations of the NP-ZVI doses. The peak values of the TCOD reduction efficiency

coincided with the higher $CH_4$ daily production of 0.015, 0.011, and 0.008 $Nm^3$ $CH_4$/kg vs. 30 mg NP-ZVI/g VS at MVR-4:1, MVR-2:1, and MVR-6:1, respectively. The performance of the E10 is notable for being the worst among the tests with NP-ZVI addition. As a result, we concluded that the concentration exceeded 30 NP-ZVI mg/g VS and that MVR-8:1 and MVR-10:1 had a negative effect on the hydrolysis percentage.

The hydrolysis percentages were calculated to determine whether the increased SCOD was caused by an increased hydrolysis rate or SCOD accumulation due to a slower rate of methanogen consumption. Therefore, we concluded that, at specific doses, NP-ZVI had an enhanced effect on the hydrolysis stage. MVR-4:1 achieved maximum dye adsorption capability in 5204, 3628, 2615, 2290, and 2016, 1845 mg/g of 30, 20, 10, 50, 60 mg NP-ZVI/g. VS. The strength of the catalyzed reaction and the subsequent interaction of the moles of $Fe^{3+}$ and $Fe^{2+}$ with $BH^{4-}$ as a result of its particular structure may be related to the stoichiometric chemistry. The size of the MVR-4:1 particle was in the range of 20–56 nm (90% of the particles had a size less than 30 nm), the surface area value was 24.6–31.7 $m^2$/g, and the surface thickness was in the range of 2–5 nm. The results revealed that the 30 mg NP-ZVI/g VS test at MVR-4:1 improved the hydrolysis percentage to 86.67%. The medium aqueous reactions were rapid at 40 °C, and no oxidation occurred in the substrate mixture; these results are consistent with those reported by Shubair et al. [51]. This may be attributed to the strength of the catalytic reaction generated by the specific surface area of NP-ZVI and the use of N2 gas as the ambient for AcoD in this investigation.

According to Pulline et al. [52], for adsorption and reduction, NP-ZVI was preferred due to its large surface area and its possession of more reactive surfaces than micro- or large-sized particles. With a dosage of 10 g/L of ZVI (0.2 mm), Zhao et al. [53] showed a significant increase in activated sludge substrate protein hydrolysis. Additionally, they have shown an increase in the activity of the glycosidase and protease enzymes by 27% and 63%, respectively. At a dosage of 27 g/L, the soluble polysaccharides and protein increased by 11.6 and 31.2%, respectively [54]. Zhang et al. [55] found that one g/L ZVI increased the polysaccharide degradation by 15.8% compared to the control. In addition, they reported that the surface area of iron nanoparticles is relatively large compared to their size, which led to the abundance of ionized iron ions ($Fe^{+3}$) in more metabolic pathways and increased the hydrolysis rate. According to Zhao et al. [53], the hydrolysis rates can be enhanced by the intermolecular interactions of Columbus. This is achieved by luring the negatively charged cations present in complex molecules to the positive magnetite nanoparticles.

### 3.1.2. Effects of Dose and Size of NP-ZVIs on the Percentage of Acidification

The acidogenic microbes broke down the substrate mixture to produce VFA (acetate, butyrate, and propionate). The acetate concentration was tracked during the AcoD to evaluate the acidification percentage. When the substrate was acidified, VFAs were produced that affected $CH_4$ production. The effect of NP-ZVI's addition on the VFAs' potential for the synthesis of methane was investigated. Acetate production was significantly stimulated by B30, C30, A30, and B20, as illustrated in Figure 2, with maximum levels of 9568 and 9489 mg/L relative to 2337 mg/L for the control test after day six of the incubation period. While the acetate production rates for all tests containing 40–50 mg NP-ZVI/g VS were the lowest of all tests containing NP-ZVI, particularly the E50 test, tracking the behavior of acetate concentration over the first six days revealed that all of the tests with the addition of NP-ZVI had high acetate concentration production, which then decreased and disappeared after day 15. The best concentration in terms of the acidification percentage was 30 mg NP-ZVI/g VS and MVR-4:1, which corresponded to the largest surface area of all the NP-ZVI particles. In addition, the studies with the addition of more than 30 mg of NP-ZVI revealed low TCOD removal and $CH_4$ production (after day one), followed by increased TCOD removal but low $CH_4$ production (after day six), and finally, again, with lower TCOD removal but increased VFA and $CH_4$ production (after day 10). This pattern can result from the substrate's sorption during high COD removal (exceeding the intended OLR while applying an iron overload). In agreement with the studies conducted by [53,56],

these results are similar. However, the results obtained in this investigation do not agree with those obtained by Zhang et al. [57] as they found that adding 50 mg/L NP increased H2 production by 1.2-fold compared to the control test.

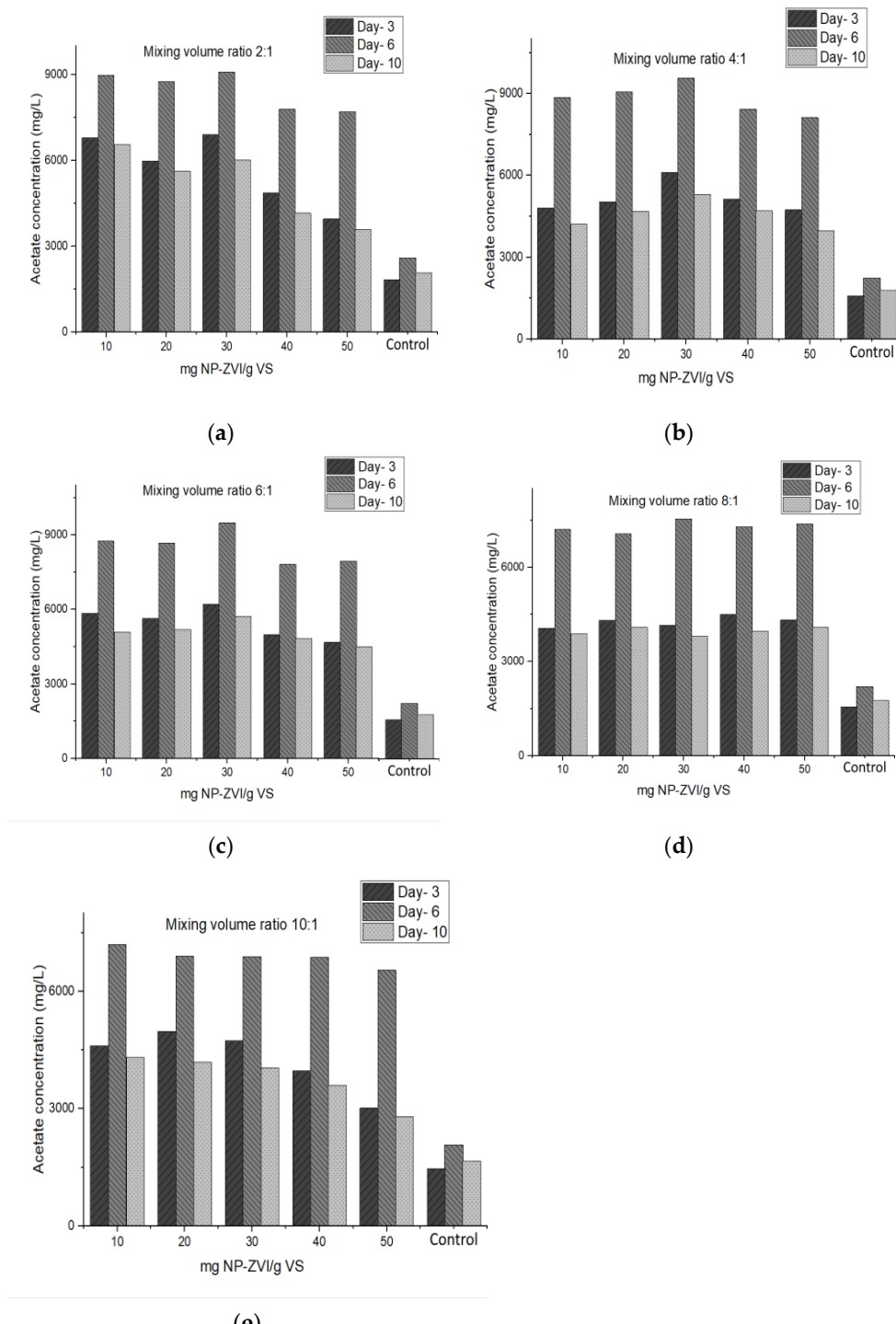

**Figure 2.** (**a**) Effect of different doses of NP-ZVI with MVR-2:1 on acetate concentration; (**b**) Effect of different doses of NP-ZVI with MVR-4:1 on acetate concentration; (**c**) Effect of different doses of NP-ZVI with MVR-6:1 on acetate concentration; (**d**) Effect of different doses of NP-ZVI with MVR-8:1 on acetate concentration; (**e**) Effect of different doses of NP-ZVI with MVR-10:1 on acetate concentration.

In general, VFA production is strongly correlated with the dose of NP-ZVI, as shown in Figure 2. A simultaneous decline in $CH_4$ production was observed with the increased

NP-ZVI doses over 40 mg NP-ZVI/g VS. Consequently, the VFA content declined simultaneously with the $CH_4$ production rate, which increased significantly after six days. Calculations were conducted to assess the net increase in the VFA production induced by NP-ZVI and the acidification percentage, which includes the $CH_4$ content (VFA consumption) and VFA generation. Positive effects were observed in the acidification stage and when increasing the acidification percentage; B3 achieved the highest increase in the acidification percentage, of 51.3%, followed by A30, C30, and B20 at 48.7%, 46.9%, and 44.3%, respectively, while E50 achieved the lowest increase of 10.2%.

Various studies have reported that generated VFAs outperform traditional carbon sources in terms of enhancing the removal of phosphate and nitrogen [55,58,59]. The results showed that at low concentrations of NP-ZVI, when nitrogen was limited, the generated VFA increased lipid synthesis, which improved the microorganism activity. Additionally, there was no correlation between the pace at which the acetogenic and methanogenic archaea degraded the VFAs and the rate at which the acidogenic bacteria produced them; these results are in line with those reported by Aboudi et al. [60]. There are several parameters that affect the generation and consumption of acetate. Most of them are associated with acidogenesis, acetogenesis, syntrophic acetate oxidation, and methanogenesis. All of them play a role in AcoD. As acetate can serve as a methanogenic precursor and as an intermediate product, it is pertinent to thoroughly investigate its role in AcoD. Acetate was the primary product in the fermentation of amino acids-glycerol and polysaccharides into peptides when the BMP was run steadily under low $H_2$ partial pressure. Most of the substrate mixture was converted to acetate and $H_2$ instead of reduced products, such as propionate, ethanol, and butyrate. These results agree with those reported in some of the literature [58,61].

3.1.3. Effects of Dose and Size NP-ZVI on the Removal Contaminant and Biologica Treatment

According to some studies, a thin surface with a thickness of 2–4 nm, primarily composed of iron oxide or amorphous oxide, will be formed at the clumped surface and between the particles [55,57–60]. The chemical and physical (surface area and thickness) features of the NP-ZVI additions greatly aided the microorganism's enzymatic activity for biogas and CH4 production and catalytically enhanced the substrate treatability efficiency. Figure 3 shows the removal efficiency of the contaminant from the substrate mixture after an incubation period. Based on the results, the addition of NP-ZVI increased the reactivity for the rapid elimination of contaminants.

As shown in Figure 3, significant removal efficiencies were noticed for the TS, VS, and contaminant. According to the MVR of $NaBH_4$:$FeSO_4$.$7H_2O$, the highest removal effectiveness of the TS, TCOD, color, and turbidity in the tests was 4:1 > 6:1 > 2:1 > 8:1 > 10:1, respectively, and 4:1 > 2:1 > 6:1 > 8:1 > 10:1 for the VS removal efficiency. While the performance of the different pretreatment doses of NP-ZVI varied greatly at the same MVR of $NaBH_4$:$FeSO_4$.$7H_2O$, the significant effect was 30 mg NP-ZVI/g VS for the MVR-4:1, MVR-6:1, MVR-8:1, and MVR-10:1, and 10 mg NP-ZVI/g VS for the 2:1 mixing volume ratio. These removal efficiencies were reflected in the methane content and AcoD stability for all four stages. The results generally revealed that the stimulating performance and contaminant removal efficiency of MVR-8:1 and MVR-10:1 were insignificant. Among the COD components (indigestible and digestible), only SCOD is utilized in the metabolism of microorganisms.

The presence of NP-ZVI in all of the tests improved the process of active bio-conversion by an up to 82% removal rate with fewer residual SCOD, while the control test (NP-ZVI free) removal rate was 47.13%. These results are consistent with previous research, which found that specific concentrations of nanoparticles-ZVI supplements enhanced the substrate mixture degradation in BMP tests. According to Zhao et al. [53], at 10 g/L ZVI (diameter of 0.2 mm), a 30–35% $CH_4$ improvement was achieved, which was more than in the control test, and the NP-ZVI increased up to 32.3% of the co-enzyme F420 activity and enhanced the

syntrophic metabolism between the methanogens and syntrophic bacteria. Wu et al. [62] reported that 25 g ZVI/L (38 μm of diameter) of wastewater led to an 89.2% improvement in the removal efficiency of the contaminants. In the case of Cai et al. (2018) [63], 28 g ZVI/L (0.2 mm of diameter) of excess sludge increased the TCOD removal by 178.4% compared to the control test.

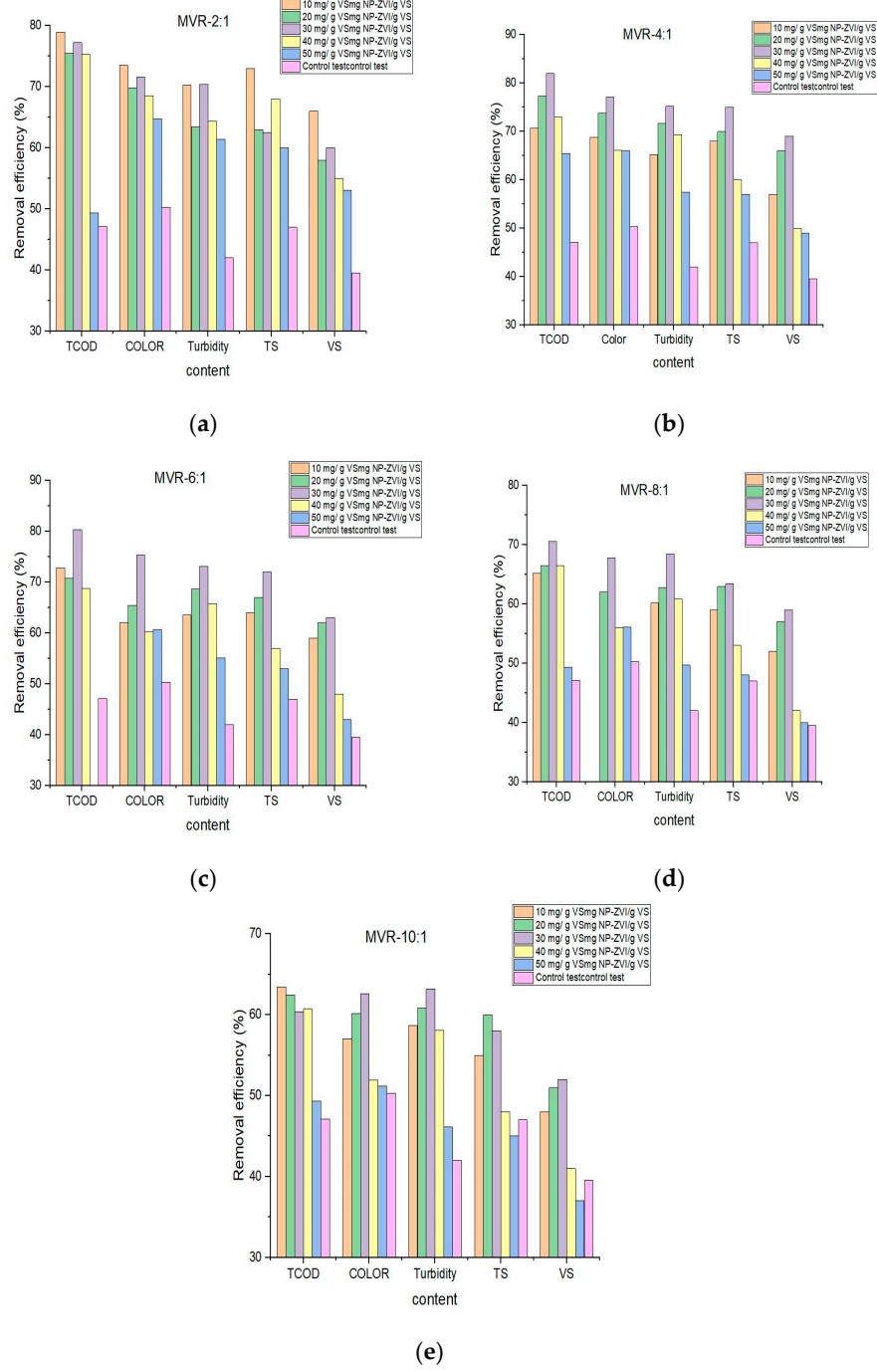

**Figure 3.** (**a**) Effect of NP-ZVI concentration and MVR-2:1 of NaBH$_4$:FeSO$_4$.7H$_2$O on TS, VS, and contaminant removal efficiency; (**b**) Effect of NP-ZVI concentration and MVR -4:1 of NaBH$_4$:FeSO$_4$.7H$_2$O on TS, VS, and contaminant removal efficiency; (**c**) Effect of NP-ZVI concentration and MVR-6:1 of NaBH$_4$:FeSO$_4$.7H$_2$O on TS, VS, and contaminant removal efficiency; (**d**) Effect of NP-ZVI concentration and MVR-8:1 of NaBH$_4$:FeSO$_4$.7H$_2$O on TS, VS, and contaminant removal efficiency; (**e**) Effect of NP-ZVI concentration and MVR-10:1 of NaBH$_4$:FeSO$_4$.7H$_2$O on TS, VS, and contaminant removal efficiency.

In the generation and transport of electrons, removal efficiency plays a role. Contaminants obtain the electrons produced by NP-ZVI, which transform them into less hazardous or non-toxic compounds. Microorganisms' metabolic processes are aided by the electrons produced by the oxidation of NP-ZVI in the environment. It can generate $H_2O_2$ by using the available oxygen to transfer electrons into $O_2$. Extremely high doses of more than 30 mg NP-ZVI/g VS particle showed only a slight improvement in contaminant removal. In addition, due to its immediate reactivity with the aqueous substrate, it was unable to complete a timely recovery attempt. However, a recent report showed that a microscale of ZVI at high doses has a low effect on the AD of some waste substrates [64]. This demonstrates that, when compared to a control reactor (without additions), metal-based oxides' enzymatic and catalytic properties promote microbial degradation [23,37]. This conclusion is in line with recent studies [47,53,57]. These studies observed that organic fraction waste treatment with a particular level of iron nanoparticles increased the removal efficiency of the contaminants versus the control test. Additionally, the findings support research that suggested NP-ZVI additions to the AD can minimize the bioreactor's lag phase while enhancing the treatment effectiveness [53].

### 3.1.4. Effects of Dose and Size of NP-ZVI on the Biogas Yield and Methane Content

Tables 4 and 5 display the impacts of the NP-ZVI concentration on the biogas yield and $CH_4$ content at different MVRs. The highest biogas production rate (first peak) was observed for all tests after six days. The second peak of biogas production in the substrate mixture with additions of 30 mg NP-ZVI/g VS, 20 mg NP-ZVI/g VS, and 10 mg NP-ZVI/g VS was reached after 14 days, 16 days, and 15 days, respectively, of fermentation. This result shows that co-digestion occurs and that OMWW:CM is more readily and rapidly metabolized when NP-ZVI concentrations of 30 mg NP-ZVI/g VS are present. After that, the generation rate decreased daily until the experiment's termination on day thirty. The addition of 30 mg NP-ZVI/g VS in the digestion of OMWW:CM resulted in biogas and $CH_4$ contents in the range of 80–10.2% and 22.86–16.95%, respectively, at various MVR. Furthermore, the addition of NP-ZVI $\leq$ 30 mg and different MVRs enhanced the daily production of biogas and the $CH_4$ content. The maximum biogas yield among the tests with the addition of NP-ZVI doses was 0.389 $Nm^3$/kg VS for the B test (30 mg NP-ZVI/g VS and an MVR-4:1), with a $CH_4$ content of 68.34%, while the yield of biogas and the $CH_4$ content of the control test was 0.216 $Nm^3$/kg VS and 55.62%, respectively.

**Table 4.** Biogas yield of all BMP tests of substrates mixture of OMWW:CM with different doses of NP-ZVI and various MVR (biogas yield of the control test was 0.216 $Nm^3$/kg VS).

| Mixing Volume Ratio (NaBH$_4$:FeSO$_4$.7H$_2$O) | Biogas (Nm$^3$/kg.VS) | | | | |
|---|---|---|---|---|---|
| | 10 mg/g VS | 20 mg/g VS | 30 mg/g VS | 40 mg/g VS | 50 mg/g VS |
| 2:1 | 0.311 ± 0.015 | 0.28 ± 0.051 | 0.282 ± 0.023 | 0.182 ± 0.011 | 0.128 ± 0.002 |
| 4:1 | 0.32 ± 0.046 | 0.335 ± 0.013 | 0.389 ± 0.005 | 0.158 ± 0.004 | 0.107 ± 0.019 |
| 6:1 | 0.264 ± 0.027 | 0.305 ± 0.01 | 0.356 ± 0.07 | 0.136 ± 0.013 | 0.083 ± 0.008 |
| 8:1 | 0.237 ± 0.014 | 0.23 ± 0.021 | 0.245 ± 0.022 | 0.105 ± 0.034 | 0.064 |
| 10:1 | 0.222 ± 0.006 | 0.228 ± 0.015 | 0.238 ± 0.019 | 0.106 ± 0.009 | 0.055 ± 0.015 |

The daily production of biogas in the first days of each test was higher than those on the other days, but when the production was tracked, the test with NP-ZVI > 30 mg/g VS added had a smaller amount of CH4 compared to $CO_2$. Compared to the biogas and $CH_4$ content of the control test, the negative effect of high doses of NP-ZVI, alongside all the MVR, increased for doses of 40 mg NP-ZVI/g VS and 50 mg NP-ZVI/g VS. In all the NP-ZVI additions, the initial digestion response was observed in all the MVRs (Figures 1–3). The high doses of NP-ZVI (>30 mg/g VS) during the first three stages of AcoD also had a positive impact. However, at the last stage (methanogens), their impacts became negative as a result of the accumulation of the buffer capacity. In addition, the tests with NP-ZVI > 30

mg/g VS had the lowest $CH_4$ content. This could be explained by the anaerobic consortium being inhibited as a result of the specific interactions between ammonia, VFA, and phenolic chemicals [65].

**Table 5.** Methane content of all BMP tests of substrates mixture of OMWW:CM with different doses of NP-ZVI and various MVR (the $CH_4$ content of the control test was 55.62%).

| Mixing Volume Ratio (NaBH$_4$:FeSO$_4$.7H$_2$O) | Methane (%) | | | | |
|---|---|---|---|---|---|
| | 10 mg/g VS | 20 mg/g VS | 30 mg/g VS | 40 mg/g VS | 50 mg/g VS |
| 2:1 | 66.47 | 65.08 | 67.08 | 52.43 | 50.05 |
| 4:1 | 67.35 | 66.89 | 68.34 | 49.05 | 46.22 |
| 6:1 | 65.78 | 66.92 | 67.79 | 46.42 | 45.18 |
| 8:1 | 63.11 | 64.88 | 65.93 | 45.38 | 45.1 |
| 10:1 | 56.47 | 59.15 | 65.05 | 44.93 | 42.79 |

Meanwhile, at the lower doses (<30 mg NP-ZVI/g VS), the stimulatory performance of AcoD was insignificant, as reflected in the biogas yield and $CH_4$ in the range of 22.87–16.5%, 20.26–6.34%, and 21.09–1.53% by 30 mg/gVS, 20 mg/gVS, and 10 mg/gVS, respectively, at various MVR, versus the control test.

Additionally, sulphide compounds can be sequestered by NP-ZVI. At NP-ZVI concentrations of 10–30 mg/g VS, the interaction of sulphide with the oxide shell of nano-Fe to create FeS and some $FeS^2$ was the primary mechanism by which $H_2S$ was eliminated in AcoD. This reaction causes the concentration of $H_2S$ in the biogas that is generated to decrease significantly; this agrees with [66]. Another advantage of NP-ZVI (10–30 mg/g VS) bioreduction could be eliminating humic and fulvic acids, which inhibit acidogenesis and methanogenesis. These compounds with negatively charged OH and COOH groups could have formed as a result of $Fe(OH)^+$ and $Fe^{2+}$ cations or as a result of $Fe^{2+}$ bridges forming between negatively charged clay particles and humic acid.

These compounds with negatively charged OH and COOH groups could have formed as a result of $Fe(OH)^+$ and $Fe^{2+}$ cations, or as a result of $Fe^{2+}$ bridges forming between negatively charged clay particles and humic acid; this agrees with Sun et al. [67].

Zhao et al. [56] demonstrated that 30–35% more $CH_4$ was produced with the addition of 10 g/L ZVI powder with a 0.2 mm diameter compared to a control test. According to Feng et al. [68], by adding 20 g/L of waste-activated sludge, 43.5% of $CH_4$ productivity was demonstrated. A 15 g/L ZVI addition increased the biogas production by 82% in Ibrahim and Abdulaziz [69]. This result contradicts that of the present investigation, and thus the dosage of 30 mg NP-ZVI/g VS (0.468 g/L) was set at MVR-4:1 in the extended studies. Smaller particle sizes of NP-ZVIs were shaped from MVR-4:1, MVR-2:1, and MVR-6:1, in ascending order. As a result of the smaller particle sizes, the corrosion hydrogen generation rates increased, which demonstrated a more substantial effect on AcoD, which is in line with Dong et al.'s [65] findings. This study showed that high concentrations of NP-ZVI resulted in lower solubilization (see Figure 1a–c), which led to excessive acidification. The NP-ZVI particles were encapsulated by graphitic carbon layers, and the same results were reported by [14,15]. Furthermore, high doses of NP-ZVI can inhibit bacterial activity by causing membrane damage via the reductive breakdown of enzymatic reactions and functional groups. These effects may be related to the highly catalytic conditions established at the compound's surface. As a result, the hydrogen synthesis and concentration will be faster, leading to the build-up of VFAs. These ultimately resulted in methanogenesis inhibition, which is consistent with the results reported by [35–37].

These results show that microbes adapted to the feedstock were enhanced by the large surface area and the suitable concentration of NP-ZVI. Additionally, it has been demonstrated that using this technique of NP-ZVI as an enhancement of AcoD can improve the mass transfer and syntrophic cooperation between acidogenic and methanogenic microbes. Knowing that methanization is inhibited and the acidification stage is delayed when OMWW:CM and high concentrations of NP-ZVI are co-digested, it is obvious that

employing modest concentrations and sizes of NP-ZVI reduces the co-harmful substrate's effects on AcoD and increases the $CH_4$.

The microorganisms may have been inhibited or poisoned as a result of a specific concentration and size of NP-ZVIs, resulting in an insignificant performance [21–23]. This enhanced the interaction between biogas production and methanogenic activity. Despite the fact that the biogas production during AcoD with NP-ZVI > 30 mg/g VS drastically decreased, the fall continued over time. This phenomenon may be explained by the stimulating effects of the concentration and size of NP-ZVI on the bacterial adhesion and metabolic co-enzymes necessary for the production's hydrolysis, acidification, and methanation stages of the AcoD of biogas [21,30]. NP-ZVIs have a significant impact on microorganism receptor activation and interaction for methanogenesis and subsequent nutritional enhancement, according to Kassab et al. [21].

### 4. Conclusions

The influence of the NP-ZVI concentration and mixing volume ratio of NaBH4:FeSO4.7H2O (effect on particle size) on the performance of the AcoD process of OMWW:CM co-digestion in BMP conditions was investigated in mesophilic conditions. The AcoD process was improved by adding NP-ZVI $\leq$ 30 mg/g VS, while the higher concentration had a negative impact on the AcoD. The $CH_4$ content was improved by a range of 22.87–16.5%, 21.09–1.53%, and 20.26–6.34%, by 30, 10, 20 mg NP-ZVI/g VS, respectively, at MVR-4:1, MVR-6:1, MVR-2:1, MVR-8:1, MVR-10:1, ascending, versus the control test. The highest biogas yields, methane content, and biodegradability were obtained with a concentration of 30 mg NP-ZVI/g VS at MVR-4:1 to AcoD, providing the best treatability performance. The removal efficiency of the TCOD, turbidity, and color were enhanced to 73.99%, 79.07%, and 53.41%, respectively, while the TS and VS reductions were enhanced by 5.57 and 74.41%.

In addition, it enhanced the hydrolysis percentage up to 86.67% with respect to the control test.

The acidification was also greatly enhanced, with acetate being the most prevalent VFA. In the incubated tests, the NP-ZVI concentrations of 30 mg/g VS with MVR-4:1, MVR-2:1, and MVR-6:1 achieved the highest acidification percentages, which were enhanced by up to 51.3%, 48.7%, and 44.3%, respectively, compared to control incubation.

**Author Contributions:** The author K.A.B.A., the executor and designer of the campaign, wrote the paper and elaborated on the data, also, biogas process. K.T., K.A.B.A., R.A.D. and A.K. analyzed the NP-ZVI behavior, and BMP analysis. K.K.A.-Z., A.A.B. and E.E.Y. conducted experiments, prepare a framework. L.A.A.-S. and K.T. reviewed and data analysis. The author Y.D. and K.T., J.A.A.-T. and B.M.A. prepared and analyzed data, results analysis, and review. K.A.B.A., J.A.A.-T. and B.M.A. editing according to the reviewer comments. All authors have read and agreed to the published version of the manuscript.

**Funding:** This research has been carried out with the financial assistance of the Al-Balqa Applied University with project-Grant No. DSR-2022#477 and European Union under the ENI CBC Mediterranean Sea Basin Programme, Project B_A.2.1_0088_MEDQUAD. The living lab. Smart Water Use Applications (SWUAP) was used to conduct the test of specimens. In addition, the authors thank UN for contribution the funding through Perez-Guerrero Trust Fund for South-South Cooperation.

**Data Availability Statement:** All data included in the manuscript.

**Conflicts of Interest:** The authors declare there is no conflict of interest. The authors also declare that the research was conducted in the absence of any commercial or financial relationships that could be construed as a potential conflict of interest.

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
