# Peer review of "The Effects of Nanoparticles- Zerovalent Iron on Sustainable Biomethane Production through Co-Digestion of Olive Mill Wastewater and Chicken Manure"

_fermentation, doi:10.3390/fermentation9020183_

Round 1
Reviewer 1 Report
Corrections are noted in the comments of the attached file.

Author Response
I am attaching a cover letter with responses to the referee point by point.

Reviewer 2 Report
The article describes the preparation of zero-valent iron nanoparticles (NP-ZVI) and their use as an adjuvant in the process of anaerobic co-digestion of olive-mill wastewater (OMW) and chicken manure (CM). Different contents of NP-ZVI prepared from different mixing volume ratios (MVR) were used. The results indicate that NP-ZVI improve the performance of the co-digestion process, especially for NP-ZVI ≤ 30 mg/g VS at MVR 2:1, 4:1 and 6:1. NP-ZVI contents above 30 mg/g VS have some detrimental effects on the process.
Comments:
Some flaws in the wording make some sentences incomprehensible (e.g. lines 55-56; “However” on lines 65-66; line 82; line 184; line 216; line 294; line 314; line 336; line 468; lines 551- 553 what is “respectively” referring to?).
Lines 144-146 are unnecessary because the method of determining solids is common knowledge.
Line 150, the equation number is missing.
Line 199, correct for “equation 4”.
Section 2.4: the definition of MVR and how they were implemented, as well as their relevance are unclear.
Line 212, completely incomprehensible, please correct.
Lines 244-245, clarify whether the reactors used were batch, if they were, it makes no sense to speak of hydraulic retention time.
Please correct spelling errors in the header of Table 2.
Line 255, use mg/L and not mg/l.
Line 285, the results discussed in the following lines are not explicit in Table 3, nor can they be verified.
Lines 299-300 please clarify what is “hydrolysis concentration”?
Line 296, reconsider the use of the word “versus”.
Figure 1, axis YY, reconsider name of the axis, is it g/L substrate? What is “substrate” in this case? Or do the authors mean “reaction volume”?
Lines 353-354, was the TCOD determined for the liquid phase only? How do the authors explain that TCOD removal is not accompanied by CH4 production?
Line 409, what is the “contaminant”?
Figure 3, the YY axes must always have the same scale (100%, as these are percentages).
Line 448-449, are the amounts shown the removals or increases in removals?
Line 476, verify if the CH4 content values ​​are correct, they do not match those in Table 4.
Section 3.5: Consider presenting methanization efficiency results that allow the biological removal of substrates to be assessed.
Author Response

(The authors gave the same response as above.)

Round 2
Reviewer 1 Report
The authors complied with most of the suggested corrections, but English is still very difficult to understand in several parts of the manuscript. The manuscript needs extensive editing of English language and style required. Also, many corrections placed in the Author response were not considered in the manuscript (some are highlighted in the attached file).

Author Response
Here is attached file with point-by-point responses to the reviewer’s comments.
Reviewer 2 Report
I consider that my previous comments were dully addressed.
Author Response
Dear Editor,
We appreciate you and the reviewers for your precious time reviewing our paper and providing
valuable comments. It was your valuable and insightful comments that led to possible
improvements in the current version. The authors have carefully considered the comments and
tried our best to address every one of them. After careful revision, we hope the manuscript will
meet your high standards.